# The Influence of Tyrosol-Enriched *Rhodiola sachalinensis* Extracts Bioconverted by the Mycelium of *Bovista plumbe* on Scopolamine-Induced Cognitive, Behavioral, and Physiological Responses in Mice

**DOI:** 10.3390/molecules27144455

**Published:** 2022-07-12

**Authors:** Mi-Jin Kwon, Ju-Woon Lee, Kwan-Soo Kim, Hao Chen, Cheng-Bi Cui, Gye Won Lee, Young Ho Cho

**Affiliations:** 1Division of Efficiency Evaluation of Biomolecules, PSA Co., Ltd., Pusan 48513, Korea; kwonmj1108@naver.com (M.-J.K.); sjwlee@naver.com (J.-W.L.); 2Greenpia Technology Inc., Yeoju-si 12619, Korea; kwanskim9@nate.com; 3Suite 18B Sea View Plaza, 18 Tai Zi Road Shekou, Shenzhen 518067, China; haochen@siae.cn; 4Key Laboratory of Changbai Mountain Biological Resources and Functional Molecular Education, Yanbian University, Yanji 133002, China; cuichengbi@ybu.edu.cn; 5Department of Pharmaceutics and Biotechnology, Konyang University, Daejeon 35365, Korea; pckmon@konyang.ac.kr

**Keywords:** Alzheimer’s disease, *Rhodiola sachalinensis*, bioconversion, antioxidant, Nrf2/HO-1

## Abstract

Alzheimer’s disease (AD) is an age-related neurodegenerative disorder characterized by cognitive deficits, which are accompanied by memory loss and cognitive disruption. *Rhodiola sachalinensis* (RSE) is a medicinal plant that has been used in northeastern Asia for various pharmacological activities. We attempted to carry out the bioconversion of RSE (Bio-RSE) using the mycelium of *Bovista plumbe* to obtain tyrosol-enriched Bio-RSE. The objective of this study was to investigate the effects of Bio-RSE on the activation of the cholinergic system and the inhibition of oxidative stress in mice with scopolamine (Sco)-induced memory impairment. Sco (1 mg/kg body weight, i.p.) impaired the mice’s performance on the Y-maze test, passive avoidance test, and water maze test. However, the number of abnormal behaviors was reduced in the groups supplemented with Bio-RSE. Bio-RSE treatment improved working memory and avoidance times against electronic shock, increased step-through latency, and reduced the time to reach the escape zone in the water maze test. Bio-RSE dramatically improved the cholinergic system by decreasing acetylcholinesterase activity and regulated oxidative stress by increasing antioxidant enzymes (superoxide dismutase (SOD) and catalase (CAT)). The reduction in nuclear factor erythroid 2-related factor 2 (Nrf2)/heme oxygenase-1 (HO-1) signaling in the brain tissue due to scopolamine was restored by the administration of Bio-RSE. Bio-RSE also significantly decreased amyloid-beta 1–42 (Aβ1–42) and amyloid precursor protein (APP) expression. Moreover, the increased malondialdehyde (MDA) level and low total antioxidant capacity in Sco-treated mouse brains were reversed by Bio-RSE, and an increase in Nrf2 and HO-1 was also observed. In conclusion, Bio-RSE protected against Sco-induced cognitive impairment by activating Nrf2/HO-1 signaling and may be developed as a potential beneficial material for AD.

## 1. Introduction

Memory impairment is known to be caused by severe damage to the cholinergic system, which is related to various dysfunctions in memory, learning, language, administrative function, social cognition, and motor ability [1,2]. In addition, the cognitive impairment seen during aging is caused by the accumulation of lipid peroxides or oxidative stress that accompanies the aging process, which not only reduces biological functions, but is also important in the pathogenesis of neurodegenerative diseases, such as Alzheimer’s and Parkinson’s disease [2,3].

The brain is responsible for memory and cognitive functions; thus, behavioral abnormalities and cognitive decline are linked to oxidative stress [4]. The overproduction of reactive oxygen species (ROS) leads to neuron damage in the hippocampus, resulting in the impairment of cognitive and memory functions. Furthermore, the toxicity of amyloid *β* (A*β*) is related to increased oxidative stress and lipid peroxidation in neurons [5,6]. Hence, the overproduction of ROS contributes to the progression of Alzheimer disease (AD), which causes oxidative damage via a neurotoxicity mechanism and induces memory loss [4,7,8].

As it is known that increased acetylcholine (ACh) activity helps to improve cognitive function in AD, many drugs that can enhance the function of ACh neurons according to various mechanisms of action have been developed, such as agonists for muscarinic ACh receptors, ACh production promoters, and acetylcholinesterase (AChE) inhibitors [9,10]. However, the developed drugs are not only toxic with serious side effects, but they also have a temporary and weak effect [11,12,13]; hence, they are still controversial. Therefore, it is necessary to find potential materials for AD prevention and improvement that are derived from natural products with lower toxicity and fewer side-effects.

*Rhodiola sachalinensis* (RSE) has received great interest in phytochemical investigations for many years [14], and many bioactive components have been isolated from it, such as phenylpropanoids, flavonoids, and tannins. RSE has been shown to have various health-related biological activities, including anticancer activity, antioxidant activity, anti-inflammatory properties, cardioprotective effects, and neuroprotective effects [14,15,16]. Moreover, the salidroside, rosavins, and *p*-tyrosol found in this plant provide benefits for fatigue, depression, and cognitive dysfunction [14,17].

Salidroside and tyrosol are major components commonly found in plants [17]; they mainly exist in glycoside and non-glycoside forms along with phenolic hydroxyl groups [18,19]. According to previous studies, biological activity is known to increase when glycosides are converted to non-glycosides through bioconversion [20,21,22]. That is, the specific activity involved in the metabolism of dietary phenolic compounds leads to the improvement of their nutritional and health properties; in addition, their metabolites influence the extent of their absorption as aglycone-enriched products [22,23,24].

In this study, a bioconverted *R. sachalinensis* extract (Bio-RSE) was prepared using the mycelium of *Bovista plumbea* as a fermented strain. In addition, we investigated its antidementia properties by observing the effect of Bio-RSE on cognitive function and antioxidant activity in an animal model of induced memory impairment.

## 2. Results

### 2.1. Measurement of Total Polyphenol and Flavonoid Contents of Bio-RSE

Tannic acid and quercetin were used as standard materials to measure changes in the total polyphenol and flavonoid contents before and after the bioconversion of RSE. It was shown to contain 382.7 ± 0.32 μg/mL of polyphenol compounds and 47.8 ± 0.41 μg/mL of flavonoids before bioconversion, while it had 827.8 ± 0.22 μg/mL of polyphenol compounds and 82.9 ± 0.37 μg/mL of flavonoids after bioconversion, as shown in Table 1. The bioconversion of RSE using the mycelium of *Bovista plumbea* (Bio-RSE) increased the polyphenol content by about 2.16 times and the flavonoid content by about 1.73 times compared to before bioconversion (RSE).

### 2.2. Analysis of Aglycone Content Change Using HPLC

HPLC was performed to confirm changes in salidoside and tyrosol, the targeted components, in the Bio-RSE. Salidoside (glycoside), the main active ingredient of rhododendrons, has been well studied in the literature. In this study, the presence of salidoside, the main active ingredient in *Rhodiola* extract, was confirmed by a comparison with standard products.

As a result of bioconversion, the content of salidoside in the Bio-RSE decreased, while tyrosol (aglycone), a new active ingredient, was generated and increased from 190 ± 0.47 μg/mL to 540 ± 1.09 μg/mL during the bioconversion process (Figure 1).

### 2.3. Assessment of Antioxidant Activity through DPPH and ABTS Radical-Scavenging Ability

The radical-scavenging ability of DPPH and ABTS was measured to determine the antioxidant activity of Bio-RSE. It was shown that the positive control, vitamin C, at 1.52 ± 0.75 μg/mL and the Bio-RSE at 13.7 ± 0.82 μg/mL cleared 50% of DPPH radicals. In addition, it was shown that the positive control, BHA, at 1.63 ± 0.82 μg/mL and the Bio-RSE at 3.14 ± 0.23 μg/mL cleared 50% of ABTS radicals. Therefore, Bio-RSE exhibited excellent antioxidant activity similar to that of vitamin C and BHA, which have DPPH and ABTS radical-scavenging activity, respectively (Table 2).

### 2.4. Effect of Bio-RSE on the Body Weight and Brain Weight of Sco-Injected Mice

A decline in body weight and internal organ weight is a simple and sensitive index with which to determine toxicity after the injection of Sco or the treatment of samples. Here, changes in body weight and brain weight were measured in Sco-injected mice (Table 3). Table 3 shows the results of measuring the body weight and brain weight of the experimental animals raised for 3 weeks. The body weight of the control group with memory impairment was relatively low compared to that of the other groups, and the Bio-RSE intake group with memory impairment showed almost the same level as the normal group. In addition, there was no significant difference between the groups that were administered Bio-RSE; thus, there was no increase or decrease in body weight due to the differences in Bio-RSE concentration. In addition, there was no significant difference in brain weight between the experimental groups.

### 2.5. Effect of Bio-RSE on the Activity of Transaminase, Creatine, and BUN in Sco-Injected Mice

Table 4 shows the measurements of serum GOT and GTP activity used for the evaluation of hepatotoxicity in experimental animals raised for 3 weeks. The GOT concentration was somewhat higher in the experimental group that was administered Sco, which resulted in impaired memory compared to the normal group; however, the GPT concentration was almost the same in all the experimental groups, except for the DNZP group. Therefore, there was no hepatotoxicity due to Bio-RSE administration, as the GOT and GPT concentrations in the Bio-RSE groups were slightly higher than or at the same level as those of the normal group. Meanwhile, the serum creatine and BUN activity measurements used for the evaluation of renal toxicity (Table 4) showed that there was no significant difference in serum creatine concentration between the experimental groups. The concentration of BUN was highest in the control group, where memory was impaired by the administration of Sco; however, all experimental groups showed values within the normal range (8~20 mg/dL). Thus, there was no renal toxicity due to the administration of Bio-RSE and DNPZ.

### 2.6. Effect of Bio-RSE on Antioxidant Activity in Blood

The results of the analysis of antioxidant activity in blood are shown in Table 5. The SOD activity was significantly lower in the CON group administered Sco compared to the normal group, whereas that of the Bio-RSE group increased in a concentration-dependent manner and was significantly higher in the group that was administered 200 mg/kg. Catalase activity also showed similar results; thus, it is believed that the components in Bio-RSE have antioxidant activity.

### 2.7. Effect of Bio-RSE on MDA Activity in Brain Tissue

The content of MDA, an intermediate product of lipid peroxidation, present in the brain tissue of mice was measured; the results are shown in Table 6. The level of MDA in brain tissue is a good indicator to evaluate the oxidation of brain tissue; when the oxidation of brain tissue occurs by free radicals, a large amount of lipid peroxide is generated, and the final degradation product of this lipid peroxide is MDA. As shown in Table 6, the MDA content was 16.17 ± 1.18 nmol/mg protein, which increased by about 96% in the CON group that was administered Sco compared to the normal group when Bio-RSE was administered at 100 mg/kg and 200 mg/kg. In one group, the MDA content decreased by 20.1% and 32.5%, indicating 12.93 ± 1.09 and 10.91 ± 0.09 nmol/mg protein, respectively. In the DNPZ administration group, it decreased by 15.4% compared to the Sco administration group, to 13.69 ± 1.19 nmol/mg protein.

### 2.8. Effect of Bio-HKC on Acetylcholinesterase (AChE) Activity in Brain Tissue

AChE activity was measured by extracting the mouse brains; the results are shown in Table 6. As shown in Table 6, AChE activity was the highest in the CON group induced by Sco and the lowest in the normal group and DNPZ group. In addition, in the experimental groups where Bio-RSE was administered at 50, 100, and 200 mg/kg concentrations, it decreased in a concentration-dependent manner, and, in the 200 mg/kg administration group (108.65% ± 1.99% mg protein), it appeared the be lower than in the DNPZ administration group (115.68% ± 2.56% mg protein).

### 2.9. Improvement of Memory and Learning Ability in a Y-Maze Experiment

A Y-maze experiment was conducted to confirm Bio-RSE’s effects on the improvement of spatial learning and short-term memory in memory-decreased mice treated with SCOP. The spontaneous alteration behavior of the normal (NOR) group was 68.5% ± 1.5%; that of the CON group, whose memory was impaired due to Sco administration, was 50.7% ± 1.8%, a decrease of 74.1% compared to the NOR group, indicating a significant difference (*p* < 0.05). This showed that short-term memory was impaired; thus, a model of memory impairment due to Sco was well formed (Figure 2).

In the Bio-RSE 50 group, Bio-RSE 100 group, and Bio-RSE 200 group—administered Bio-RSE at 50, 100, and 200 mg/kg, respectively—the spontaneous altering behavior was 54.7% ± 1.5%, 58.8% ± 1.7%, and 61.2% ± 1.9%, respectively, which indicated recovery to the levels of 79.9%, 85.8%, and 89.3% of the NOR group. In addition, the DNPZ group treated with donepezil, which was used as a positive control group, had a spontaneous altering behavior of 63.1% ± 0.9%. Since there was no significant difference between the experimental groups in the total number of entries in each branch, we concluded that Bio-RSE was effective in improving memory (Figure 2).

### 2.10. Improvement of Memory and Learning Ability in a Water Maze Experiment

In order to confirm the spatial learning and long-term memory improvement effect of Bio-RSE in memory-impaired mice treated with Sco, a Morris water maze experiment was conducted. As the learning progressed in all experimental groups, the time it took to find the escape zone decreased. However, in the CON group treated with Sco, the reduction time was similar to that of the first day (>60 s), which was significant compared to the NOR group. It took a great deal of time to progress. This indicated that the long-term memory impairment model was well formed, as it was judged that the escape zone could not be remembered due to Sco administration (Figure 3).

In all the experimental groups to which Bio-RSE was administered, from the third day of learning, the time taken to find the escape zone was significantly reduced compared to the CON group. It was confirmed that spatial perception was also impaired, as it decreased to 15.2 ± 2.1 s, which was 60.6% of the level of the NOR group. In contrast, in the Bio-RSE 50 group, Bio-RSE 100 group, and Bio-RSE 200 group, memory was improved, with values of 17.2 ± 1.2 s, 19.8 ± 1.4 s, and 23.1 ± 1.1 s, which were 68.5%, 78.9%, and 92.1% of that for the NOR group, respectively. In the groups administered 100 and 200 mg/kg, it was confirmed that there was a significant effect on memory improvement (Figure 3).

### 2.11. Improvement of Memory and Learning Ability in a Passive Avoidance Experiment 

A passive avoidance experiment was conducted using the characteristics of mice that prefer dark places. As a type of behavioral experiment used to evaluate memory, this method measures the time to remember an electrical stimulus. The results from the second day of the passive avoidance experiment (retention trial) are shown in Figure 4. The step-through latency of the normal control group was 121.8 ± 12.8 s; for the CON group treated with Sco, it was 58.6 ± 5.9 s. It was confirmed that the CON group treated with Sco had a reduced step-through latency compared to the NOR group, verifying the impairment of memory required for passive avoidance due to the inability to remember the electric shock the day before. In contrast, the Bio-RSE-treated groups showed a significant increase in a dose-dependent manner; the 100 mg/kg administration group had times of 76.4 ± 10.9 s and 77.6 ± 11.5 s, which were similar to those of the DNPZ administration group.

### 2.12. Effect of Bio-RSE on Aβ Levels in Serum and Brain Tissue

As shown in Figure 5, the serum Aβ1–42 level was 121.8 ± 2.08 pg/mL in the CON group treated with Sco, which was an increase of about 60.1% compared to the normal group; in the groups treated with 100 mg/kg and 200 mg/kg of Bio-RSE, compared to the control group, it decreased by 17.6% and 29.9%, with amounts of 101.3 ± 1.2 and 84.9 ± 2.09 pg/mL, respectively. In the DNPZ administration group, compared with the Sco administration group, it decreased by 17.4% to 101.2 ± 1.07 pg/mL. This pattern was also observed in the Aβ1–42 levels in brain tissue.

### 2.13. Effect of Bio-RSE on Nrf2/HO-1 and Aβ/APP Protein Expression

Next, we evaluated the effect of Bio-RSE on Nrf2/HO-1 protein expression, since the group fed Bio-RSE showed an increase in antioxidant enzymes, such as SOD and catalase, in the blood.

As expected, a Western blot analysis showed that the total expression of HO-1 and Nrf2 was higher in the groups administered Bio-RSE at 100 mg/kg and 200 mg/kg compared to the CON group. Similar to the results measured in blood, the expression of SOD and catalase in brain tissues also showed the same pattern. At the same time, Aβ1–42/APP protein expression, which affects memory impairment, was reduced in the same groups when compared to the CON group. Even in the group fed Bio-RSE at 200 mg/kg, the level was similar to that of the NOR group (Figure 6). Taken together, these results suggest that Bio-RSE is a potential Nrf2/HO-1 activator, as it appears to result in a decrease in Aβ1–42/APP due to antioxidant activity.

## 3. Discussion

The brain is responsible for memory and cognitive function, and age-related neurodegenerative disorders characterized by cognitive deficits are becoming markedly more common in the world. In fact, the exact cause of the degenerative aging disease known as Alzheimer’s disease (AD) has not been identified thus far, and an effective preventive or therapeutic agent has not been developed [25,26,27].

Behavioral abnormalities and cognitive decline in the central nervous system are linked to oxidative stress [28]. Oxidative stress occurs due to an imbalance between the formation of free radicals and a loss of the ability to detoxify these reactive molecules or to repair the damage that they cause [28,29].

Oxidative stress-induced intracellular ROS overexpression plays a critical role in inflammation development and organic DNA damage [30]. The nuclear factor E2-related factor 2 (Nrf2) transcription factor is a crucial regulator of cytoprotection and stress resistance in skin cells. In response to oxidative stress, Nrf2 is translocated to the nucleus, where it binds to antioxidant response elements (AREs) to regulate the expression of cytoprotective target genes, such as heme oxygenase-1 (HO-1), glutathione synthase (GSS), catalase (CAT), and superoxide dismutase (SOD), which attenuate mitochondrial ROS production [31,32,33]. Among them, HO-1 induction is involved in anti-inflammatory, antioxidant, and antiapoptotic effects [33].

Furthermore, the toxicity of Aβ is related to increased oxidative stress and lipid peroxidation in neurons [34]. In particular, the brain requires more oxygen to perform hippocampal synaptic functions; however, it is vulnerable to oxidative stress [34,35]. Hence, the overproduction of ROS contributes to AD progression, which causes oxidative damage via a mechanism of neurotoxicity and induces memory deficits [34,36]. Recent experiments have investigated the possible mechanism of antioxidant-based therapeutics in AD and their free-radical-scavenging activity based on increasing antioxidant enzymes from the Nrf2/HO-1 pathway and have shown that the removal of hydrogen peroxide inhibits amyloid accumulation [35]. This has been applied not only to hippocampal neurons, but also to APP/PS1 transgenic AD mice and the 3X TG mouse model, confirming the anti-AD potency effect of Nrf2-based antioxidant activity [37,38]. All of these results emphasize the protective role of Nrf2/HO-1 against the conditions of neurodegenerative diseases associated with aging and as an emerging target against oxidative stress in AD [34,35].

We found that tyrosol-enriched Bio-RSE ameliorated Sco-induced cognitive impairment in vivo. Further mechanistic studies revealed that Nrf2/HO-1 activation was involved in the neuroprotective effect of Bio-RSE, especially that rich in tyrosol, by increasing antioxidant activity and decreasing Aβ/APP accumulation.

According to a study by Marotti et al., bioactivity increases when glycosides are converted to non-glycosides through bioconversion [20,21]. In addition, it was reported that isoflavone glycoside was converted into isoflavone aglycone after 9 days in culture when soybean extract was cultured with reishi mushroom mycelium. Therefore, the generation of tyrosol in the Bio-RSE shown in this study is thought to be a metabolite remaining after the decomposition of glycoside during the bioconversion process that is used as a carbon source for the mycelium [23,39]. Furthermore, the polyphenol content increased by about 2.16 times and flavonoid content by about 1.73 times compared to before bioconversion. This is attributed to the decomposition of high-molecular-weight phenolic compounds bound to the low-molecular-weight phenolic compounds or the creation of new phenolic compounds during the process of culturing the *Bovista plumbea* mycelium used as the fermentation strain.

Salidroside and tyrosol are active ingredients extracted from *R. sachalinensis*, a plant species mostly found in mountainous areas at high altitudes in northeastern Asia that has been used in traditional medicine for a long time [14]. Reports have shown that these compounds have a wide range of pharmacological properties, including anti-hypoxia, antioxidant, antifatigue, and anti-inflammation effects, as well as improvements in brain function [14,17]. Recently, salidroside and tyrosol were shown to significantly prevent ROS-induced brain ischemic injury in vivo and reduce oxidative stress by H_2_O_2_ in vitro [18,40,41]. Tyrosol shows a stronger protective effect than salidroside, suggesting that, while both substances exhibit excellent antioxidant effects, the enhanced effect of tyrosol may be partly due to the different substituents of the glycosyl group. As it has been reported that they are closely related to the development of neurodegenerative diseases, including Parkinson’s disease and AD, by the induction of cell damage, the significant increase in antioxidant enzymes caused by tyrosol-enriched Bio-RSE administration is believed to improve memory loss caused by Sco treatment.

It is evident that oxidative stress has been recognized as a contributor to the pathogenesis of AD; that is, the severe free-radical damage due to high oxygen consumption and a lack of antioxidant enzyme availability [28,40]. It is related to protein modification induced directly by ROS or indirectly by lipid peroxidation products, which is associated with an increased risk of AD [40,41]. Accordingly, many compounds applied for the treatment or restoration of AD, such as flavonoids, melatonin, and carotenoids, including tyrosol and salidroside, have been found to possess potent antioxidant properties [4,40,42]. In this study, the activity of several main antioxidant enzymes (SOD and CAT) was markedly increased by Bio-RSE, while the level of the pro-oxidant enzyme MDA was inhibited by Bio-RSE. On the basis of these results, the inhibitory effect of lipid peroxidation in brain tissue was considered to be correlated with the antioxidant activity in the administrated Bio-RSE extract.

Consistent with other reports, cognitive dysfunction in short- and/or long-term memory and spatial learning ability was observed in Sco-treated mice during various behavioral analysis experiments, which also showed that Bio-RSE could protect against Sco-induced cognitive impairment. In the cholinergic system, choline acetyltransferase is the most important synthetic enzyme that triggers the synthesis of ACh. Acetylcholinesterase is a hydrolytic enzyme that hydrolyzes ACh rapidly. In the current study, we found that Bio-RSE inhibited the activity of AChE in Sco-treated mice and showed a similar effect to DNPZ, which was used as positive control to contrast Sco damage.

It has been reported that Sco can induce oxidative stress, resulting in neuron injury and cell death in the brains of mice. We found that the level of MDA was significantly inhibited by Bio-RSE and the antioxidant capacities of Sco-treated mice were clearly increased by Bio-RSE. Therefore, Bio-RSE could provide a neuroprotective effect against Sco-induced cholinergic system dysfunction. As expected, the expressions of Nrf2 and HO-1 were upregulated by Bio-RSE. Lastly, it was shown that Aβ1–42/APP accumulation was reduced (Figure 7).

## 4. Materials and Methods

### 4.1. Preparation of Fermented Extract of R. sacchalinensis Using Mycelium Culture and Bovista plumbea Mycelium

The mycelium of *B. plumbea* was directly obtained at a size of 5 mm, inoculated into potato dextrose agar (PDA), cultured for 7 days at 25 °C, and subcultured every 4 weeks before use. The cultured flora was cut with a cork borer (φ: 8 mm), inoculated with 5 discs in sterilized potato dextrose broth (PDB), and then cultured with shaking at 25 °C for 7 days at 100 rpm. The *R. sachalinensis* ethanol extract was prepared as a 1% solution and then autoclaved. After that, the flora cultured in PDB was inoculated with sterilized *R. sachalinensis* ethanol extract and then cultured in a shaking incubator at 25 °C for 7 days. After culturing, the mycelium of *B. plumbea* was removed from the culture medium; then, a bioconverted *R. sachalinensis* extract (Bio-RSE) was prepared.

### 4.2. Estimation of Total Phenolic and Flavonoid Content

The total phenolic content was determined by the Folin–Ciocâlteu method [43]. In brief, the extract was prepared by extracting samples for concentration 200 ppm in dimethyl sulfoxide (DMSO). About 0.1 mL sample was transferred to a 10 mL test tube, Na_2_CO_3_ 75% in 1 mL of Aquadest was added, and the solution was remixed. Folin–Cioc–lteau’s phenol reagent (1.25 mL) was added to the mixture, and the solution was remixed. After mixing, mixture incubated at room temperature for 40 min. The absorbance of the reaction mixtures was recorded at 760 nm. 

The total flavonoid content was determined by the aluminum chloride colorimetric method [43]. In brief, the extract was prepared samples for concentration 200 ppm in DMSO. About 0.5 mL of sample was transferred to a 10 mL test tube, aluminum chloride (AlCl_2_ 2%) was added (0.5 mL), and the mixture was incubated for 15 min. The absorbance of the reaction mixtures was recorded at 415 nm.

### 4.3. DPPH and ABTS Free-Radical-Scavenging Assay

Scavenging activity against the DPPH radical was assessed according to the method described by Blois [43], with some modifications. For the ABTS assay, the procedure followed the method reported by Arnao et al. [43], with some modifications.

### 4.4. Analysis of Content Changes of Salidroside and Tyrosol Using HPLC

HPLC (Shimadzu, Kyoto, Japan) was performed to confirm changes in salidroside and tyrosol present in RSE and Bio-RSE. HPLC analysis conditions were as follows: detector, PDA detector; detection wavelength, 275 nm; analytic column, TSKgel ODS-100Z (4.6 × 250 mm, 5 μm, TOSOH, Tokyo, Japan); sample injection volume, 10 μL; flow rate, 1.0 mL/min. Methanol and 5% methanol (0.04% TFA) were used as the mobile phase solvent.

### 4.5. Animals and Experimental Protocols

Male ICR mice (8 weeks old, 30–35 g) were purchased from Oriental Inc. (Seongnam, Korea). The mice were housed in plastic cages with a maintained 12 h light/dark cycle and controlled temperature (20 ± 2 °C) and humidity (50% ± 10%). The mice were given free access to water and food (5L79, Orient Inc., Seongnam, Korea). The animal protocols were confirmed by the Konyang University Institutional Animal Care and Use Committee (IACUC 21-41-A-01), and all experiments were carried out in the UNIST animal laboratory according to the protocol. The mice were acclimated for 1 week before the experiments. They were randomly divided into six groups (*n* = 12 per group): normal, control, Bio-RSE 50, Bio-RSE 100, Bio-RSE 200, and donepezil (DNPZ). The mice in the normal and control groups were orally administered 100 μL of water via an oral gauge. The mice in the Bio-RSE 50, Bio-RSE 100, and Bio-RSE 200 groups were orally administered 100 μL of Bio-RSE at doses of 50, 100, and 200 mg/kg/day, respectively. In addition, the DNPZ groups (positive control group) were orally given 100 μL of DNPZ at a dose of a 1 mg/kg/day. The normal mice were i.p. injected with 0.9% NaCl, and the other five groups were injected with scopolamine (1 mg/kg) 30 min before every behavioral experiment in the same manner.

### 4.6. Y-Maze Test

A Y-maze test was used to evaluate the ability to act sequentially and, thus, to measure short-term memory; the measuring equipment consisted of three branches, and the experiment was conducted after setting the three branches as A, B, and C, respectively. One point was given for the number of times the mouse entered each branch up to its tail and the number of times it entered each branch in turn for 10 min after inserting the mouse (actual alternation). Spontaneous alternation was defined as entering and nonoverlapping in all three and was calculated using the following equation:Change action (%) = actual change/maximum alternation × 100,
where maximum alternation is the total number of entries − 2.

### 4.7. Morris Water Maze Test

To confirm long-term memory, a Morris water maze test was performed. The test was carried out in a circular water tank, which was filled with water and mixed with nontoxic soluble black liquid gel. The water temperature was maintained at 26 ± 2 °C. The water tank was randomly divided into four quadrants, and each quadrant was marked with a different sign as a space perception cue. The hidden platform was placed at approximately 1 cm below the water surface in the center of one of the quadrants. In all experiments, SMART video tracking software 3.0 (Panlab, Barcelona, Spain) was used with a camera above the water tank to analyze mouse movements. The program was used to visually analyze mouse movements and identify significant differences for each group. Training sessions were performed over 3 days; the intention of each day was for the mice to swim and find the escape platform in 60 s using the visual clues. If the mice could not find the platform, we carefully guided the mice to the hidden platform and allowed them stay there for 20 s. On the fourth day, three trials were conducted.

### 4.8. Passive Avoidance Test

A passive avoidance test was conducted using two identical compartments (Gemini Avoidancesystem, San Diego, CA, USA), consisting of a lit and darkened compartment with an automated door in between and an electrifiable grid floor. During the acquisition phase, mice were placed in the lit compartment for familiarization for 25 s and then crossed to the darkened compartment. The door was automatically opened, and the mice received a mild electrical shock (0.3 mA, 3 s). In the retention phase, mice were again placed in the lit compartment. The latency time in the darkened compartment that was required for the mice to remain in the lit compartment was recorded as the retention time. If the mice did not enter the darkened compartment within 5 min, the latency time was recorded as 300 s. No physiological defects (i.e., motor deficits) or intrinsic cognitive impairments were observed in any of the mouse groups prior to treatment with Sco.

### 4.9. Measurement of Antioxidant Defense System in Serum

Blood was collected from the heart, left at 4 °C for 30 min, and centrifuged at 3000 rpm for 15 min (FLETA-5, Hanil Science Inc., Kimpo, Korea); then, the serum was collected and stored at −70 °C until analysis. Serum SOD activity was measured using an SOD assay kit (A001, Nanjing Bio Co., Nanjing, China). Serum CAT activity was measured using a CAT assay kit (A007-1, Nanjing Bio Co., Nanjing, China). The resulting values were expressed as a log function of the measured value.

### 4.10. Determination of Malondialdehyde (MDA) Content in Brain Tissue

After weighing the brain homogenate, it was placed in 10 volumes of 0 °C Tris buffer (pH 8.0), homogenized with a Glass-Col homogenizer, and centrifuged at 12,000 rpm for 30 min (FLETA-5, Hanil Science Inc. Korea); then, the upper layer was taken and used for the measurement of enzyme activity. After mixing 960 µL of 1% phosphoric acid with 160 µL of rat brain tissue homogenate extracted with phosphate-buffered saline, 320 µL of 0.67% thiobarbituric acid was added, and the reaction was carried out at 95 °C for 1 h. The reaction solution was centrifuged at 5000 rpm for 10 min (FLETA-5, Hanil Science Inc. Korea), and the absorbance of the supernatant was measured at 532 nm. MDA content was expressed as the concentration of nmol per mg protein [44].

### 4.11. Determination of AChE Content in Brain Tissue

After the behavioral experiments were completed, the mouse brains were removed, 10 volumes of lysis buffer was added, the mixture was homogenized and centrifuged at 12,000 rpm for 30 min (FLETA-5, Hanil Science Inc., Kimpo, Korea), and the supernatant was used to determine enzyme content. All extraction processes were performed at 4 °C. In order to estimate the protein content of the extracted enzyme solution, we used a BCA protein assay kit (Invitrogen Co., Carlsmbad, CA, USA). AChE activity in the brain tissue was expressed as the percentage activity per milligram of protein compared to the normal group [45].

### 4.12. ELISA for the Measurement of Aβ Levels

The levels of Aβ were measured using a human Aβ assay kit. This kit is a solid-phage sandwich ELISA with two kinds of highly specific antibodies, which are 70% reactive with rodent Aβ1–42. The assay was conducted following the manufacturer’s instructions (Immuno-Biological Laboratories, Gunma, Japan) as described previously [46].

### 4.13. Western Blot Analysis

Equal amounts of total prostatic protein (30 µg) were heated at 100 °C for 5 min, loaded onto 10–12% SDS-PAGE, and electrophoresed. The proteins were then transferred to an Immobilon-P polyvinylidene difluoride membrane (Millipore Corporation, Burlington, MA, USA) using a Transfer System (Bio-Rad Laboratories, Hercules, CA, USA). After blocking nonspecific binding sites with 5% nonfat milk at room temperature for 1 h, the membranes were incubated with the primary antibodies overnight at 4 °C. The immune complexes were detected with horseradish peroxidase-conjugated secondary antibodies using enhanced chemiluminescence (Amersham Pharmacia, Piscataway, NJ, USA) with exposure to an image analysis system (AI680, GE Healthcare, Uppsala, Sweden).

## 5. Conclusions

In conclusion, we successfully confirmed the neuroprotective properties of Bio-RSE and Sco-induced cognitive impairment in mice. This effect was associated with the inhibition of oxidative stress via the activation of Nrf2/HO-1 signaling. Tyrosol-enriched Bio-RSE is a potentially beneficial material that could act as a neuroprotective agent against progressive AD.

## Figures and Tables

**Figure 1 molecules-27-04455-f001:**
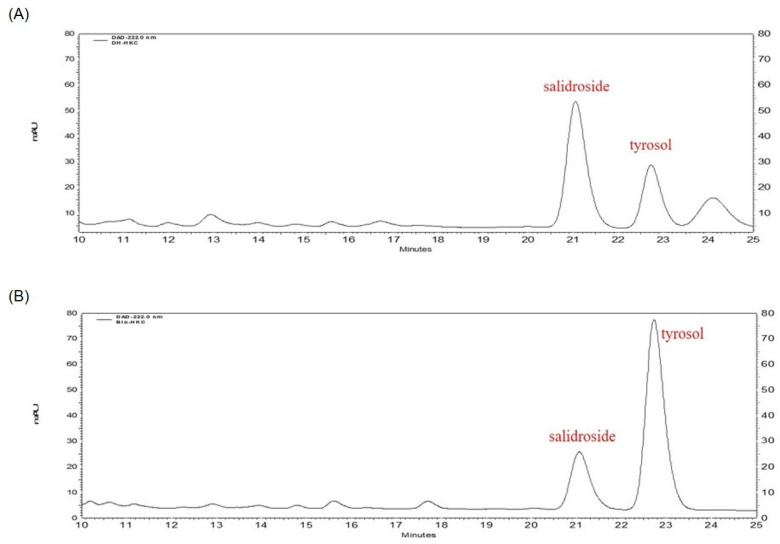
Representative HPLC chromatograms of *R. sachalinensis* extract fermented with *Bovista plumbea* (Bio-RSE) and *R. sachalinensis* extract (RSE). Chromatograms (**A**,**B**) show the contents of tyrosol and salidroside in the RSE and Bio-RSE, respectively.

**Figure 2 molecules-27-04455-f002:**
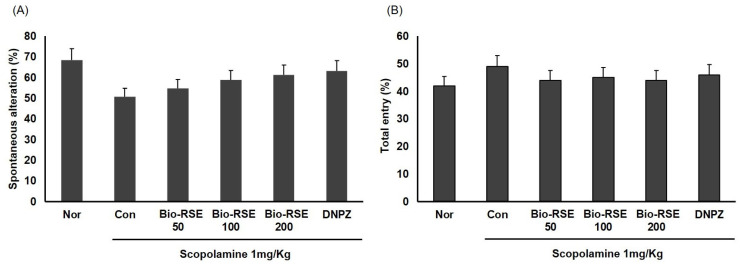
Effect of Bio-RSE on spatial alternation in the Y-maze test. Values are the mean ± SD. (**A**) The space perception abilities for the CON and Bio-RSE routes were significantly different, as determined by Student’s *t*-test (*p* < 0.05). (**B**) Total number of admissions for each experimental group. Normal—oral administration of drinking water + 0.9% NaCl i.p.; control—oral administration of drinking water + scopolamine 1 mg/kg i.p.; Bio-RSE 50—oral administration of Bio-RSE 50 mg/kg + scopolamine 1 mg/kg i.p.; Bio-RSE 100—oral administration of Bio-RSE 100 mg/kg + scopolamine 1 mg/kg i.p.; Bio-RSE 20—oral administration of Bio-RSE 200 mg/kg + scopolamine 1 mg/kg i.p.; DNPZ—oral administration of donepezil 1 mg/kg + sco 1 mg/kg i.p.

**Figure 3 molecules-27-04455-f003:**
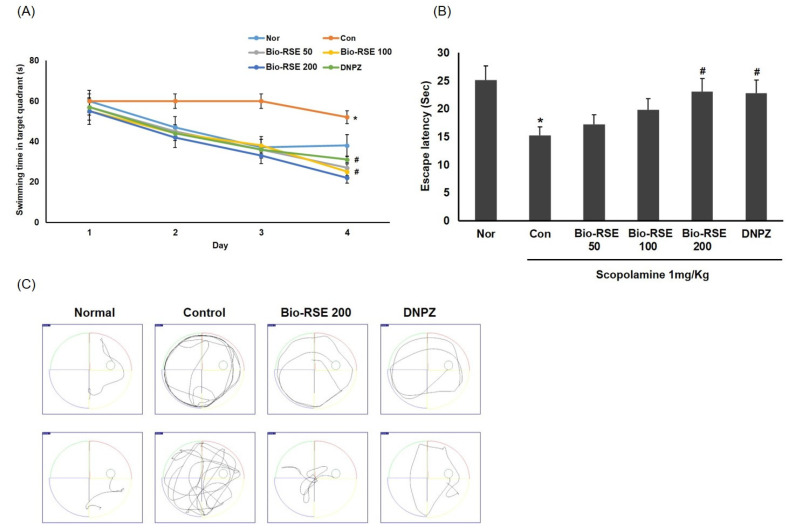
Effect of Bio-RSE on time taken to reach the platform during 4 days in the Morris water maze (**A**). Time to reach the exposed platforms on the final day in the Morris water maze test (**B**). Representative swimming tracking paths in the Morris water maze test (**C**). Values are the mean ± SD. Normal—oral administration of drinking water + 0.9% NaCl i.p.; control—oral administration of drinking water + scopolamine 1 mg/kg i.p.; Bio-RSE 50—oral administration of Bio-RSE 50 mg/kg + scopolamine 1 mg/kg i.p.; Bio-RSE 100—oral administration of Bio-RSE 100 mg/kg + scopolamine 1 mg/kg i.p.; Bio-RSE 200—oral administration of Bio-RSE 200 mg/kg + scopolamine 1 mg/kg i.p.; DNPZ—oral administration of donepezil 1 mg/kg + sco 1 mg/kg i.p. * *p* < 0.05 when compared to the vehicle-treated normal group; # *p* < 0.05 when compared to the scopolamine-treated control group.

**Figure 4 molecules-27-04455-f004:**
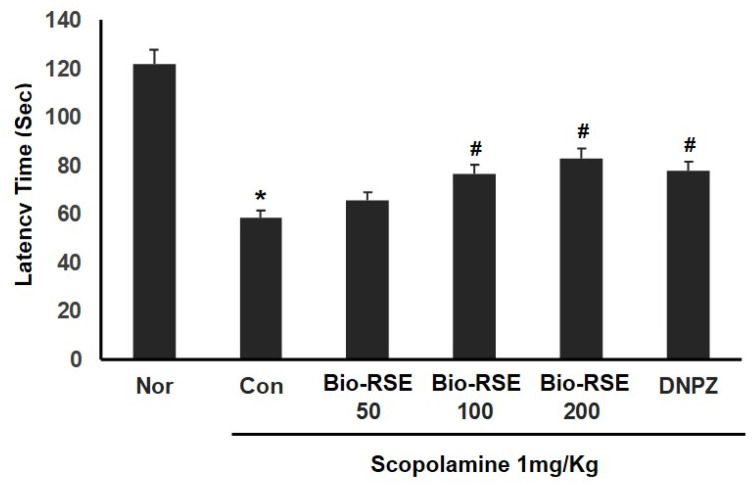
Effect of Bio-RSE on step-through latency in the passive avoidance test. Values are the mean ± SD. Normal—oral administration of drinking water + 0.9% NaCl i.p.; control—oral administration of drinking water + scopolamine 1 mg/kg i.p.; Bio-RSE 50—oral administration of Bio-RSE 50 mg/kg + scopolamine 1 mg/kg i.p.; Bio-RSE 100—oral administration of Bio-RSE 100 mg/kg + scopolamine 1 mg/kg i.p.; Bio-RSE 200—oral administration of Bio-rse 200 mg/kg + scopolamine 1 mg/kg i.p.; DNPZ—oral administration of donepezil 1 mg/kg + scopolamine 1 mg/kg i.p. * *p* < 0.05 when compared to the vehicle-treated normal group; # *p* < 0.05 when compared to the scopolamine-treated control group.

**Figure 5 molecules-27-04455-f005:**
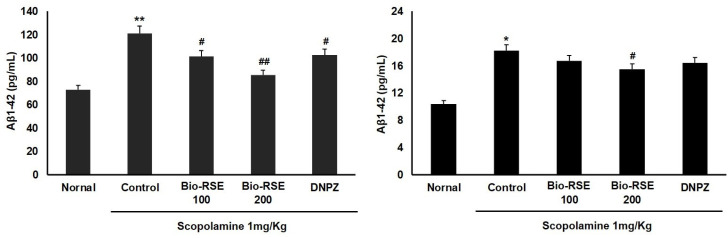
Effect of Bio-RSE on Aβ1–42 levels in serum and brain tissue. Values are the mean ± SD. Normal—oral administration of drinking water + 0.9% NaCl i.p.; control—oral administration of drinking water + scopolamine 1 mg/kg i.p.; Bio-RSE 100—oral administration of Bio-RSE 100 mg/kg + scopolamine 1 mg/kg i.p.; Bio-RSE 200—oral administration of Bio-RSE 200 mg/kg + scopolamine 1 mg/kg i.p.; DNPZ—oral administration of donepezil 1 mg/kg + scopolamine 1 mg/kg i.p. * *p* < 0.05 and ***p* < 0.01 when compared to the vehicle-treated normal group; # *p* < 0.05 and ## *p* < 0.01 when compared to the scopolamine-treated control group.

**Figure 6 molecules-27-04455-f006:**
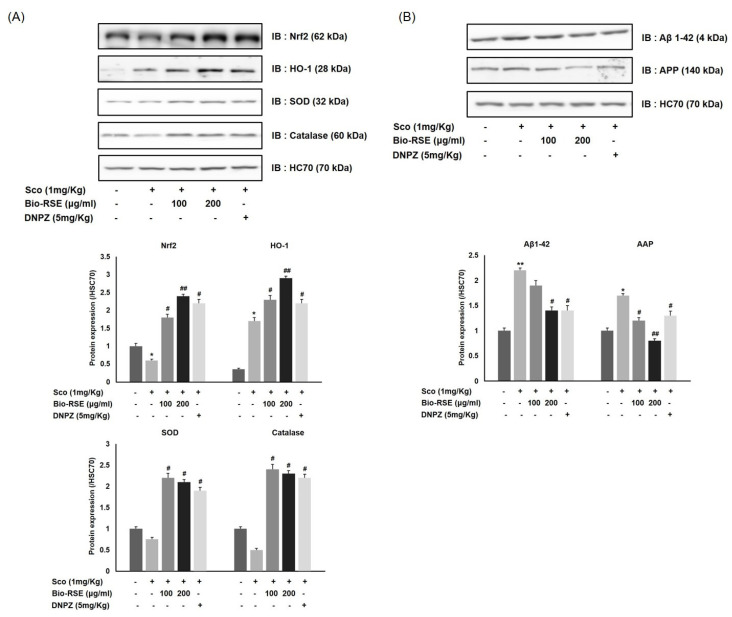
Effect of Bio-RSE on Nrf2/HO-1 protein expression (**A**) and Aβ1–42/AAP protein expression (**B**) in brain tissue. Normal—oral administration of drinking water + 0.9% NaCl i.p.; control—oral administration of drinking water + scopolamine 1 mg/kg i.p.; Bio-RSE 100—oral administration of Bio-RSE 100 mg/kg + scopolamine 1 mg/kg i.p.; Bio-RSE 200—oral administration of Bio-RSE 200 mg/kg + scopolamine 1 mg/kg i.p.; DNPZ—oral administration of donepezil 1 mg/kg + scopolamine 1 mg/kg i.p. * *p* < 0.05 and ** *p* < 0.01 when compared to the vehicle-treated normal group; # *p* < 0.05 and ## *p* < 0.01 when compared to the scopolamine-treated control group.

**Figure 7 molecules-27-04455-f007:**
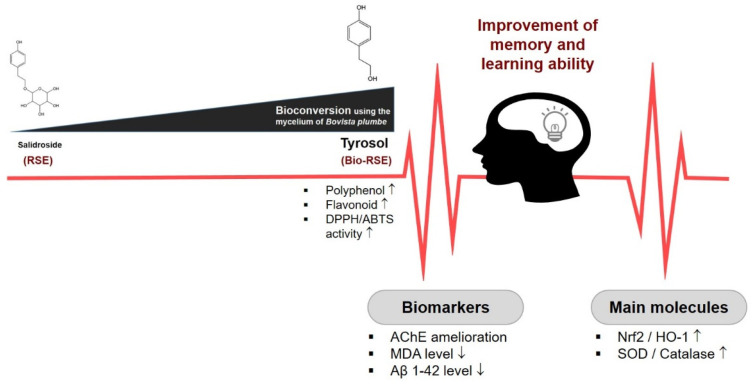
Bio-RSE could provide a neuroprotective effect against Sco-induced cholinergic system dysfunction.

**Table 1 molecules-27-04455-t001:** Changes in the content of total polyphenols and flavonoids in *R. sachalinensis* extract fermented with *Bovista plumbea* (Bio-RSE) and *R. sachalinensis* extract (RSE).

Sample	Total Polyphenol (μg/mL)	Total Flavonoid (μg/mL)
RSE	382.7 ± 0.32	47.8 ± 0.41
Bio-RSE	827.8 ± 0.22	82.9 ± 0.37

**Table 2 molecules-27-04455-t002:** DPPH and ABTS radical-scavenging activity of *R. sachalinensis* extract fermented with *Bovista plumbea* (Bio-RSE).

	DPPH (SC_50_, ug/mL)	ABTS (SC_50_, ug/mL)
Bio-RSE	13.7 ± 0.82	3.14 ± 0.23
Vitamin C	1.52 ± 0.75	-
BHA	-	1.63 ± 0.82

**Table 3 molecules-27-04455-t003:** Body weight and brain weight of mice.

Groups ^(1)^	Body Weight (g)	Brain Weight (g)
Normal	34.2 ± 0.5	0.45 ± 0.17
Control	32.7 ± 0.7	0.42 ± 0.07
Bio-RSE 50 mg/kg + Sco	33.8 ± 0.3	0.44 ± 0.10
Bio-RSE 100 mg/kg + Sco	33.4 ± 0.4	0.43 ± 0.06
Bio-RSE 200 mg/kg + Sco	34.1 ± 0.6	0.44 ± 0.04
DNPZ	33.8 ± 0.3	0.46 ± 0.05

^(1)^ Animal treatments are shown in Section 4. All values are the mean ± SD (*n* = 12/group).

**Table 4 molecules-27-04455-t004:** Serum transaminase activity, creatinine, and BUN levels in each group.

Groups ^(1)^	GOT (U/L)	GPT (U/L)	Creatine (mg/dL)	BUN (mg/dL)
Normal	39.24 ± 2.3	39.24 ± 2.3	0.62 ± 0.02	19.5 ± 1.8
Control	41.16 ± 1.3	39.26 ± 1.8	0.65 ± 0.01	22.2 ± 1.3
Bio-RSE 50 mg/kg + Sco	40.09 ± 1.7	38.16 ± 2.1	0.61 ± 0.02	20.5 ± 1.7
Bio-RSE 100 mg/kg + Sco	41.09 ± 0.8	39.11 ± 2.3	0.61 ± 0.02	19.2 ± 1.4
Bio-RSE 200 mg/kg + Sco	39.14 ± 1.1	38.38 ± 2.3	0.62 ± 0.01	19.9 ± 1.1
DNPZ	42.31 ± 1.3	41.24 ± 2.3	0.61 ± 0.02	20.5 ± 1.3

^(1)^ Animal treatments are shown in Section 4. All values are the mean ± SD (*n* = 12/group).

**Table 5 molecules-27-04455-t005:** Serum SOD and catalase levels in each group.

Groups ^(1)^	SOD (unit/mL)	Catalase (unit/g Protein)
Normal	287.84 ± 45.36	4.56 ± 0.05
Control	262.56 ± 35.26	4.21 ± 0.03
Bio-RSE 50 mg/kg + Sco	277.29 ± 40.11	4.23 ± 0.04
Bio-RSE 100 mg/kg + Sco	281.35 ± 38.39	4.32 ± 0.05
Bio-RSE 200 mg/kg + Sco	284.65 ± 36.98	4.51 ± 0.03
DNPZ	276.49 ± 41.22	4.46 ± 0.05

^(1)^ Animal treatments are shown in Section 4. All values are the mean ± SD (*n* = 12/group).

**Table 6 molecules-27-04455-t006:** Serum MDA levels and brain AChE activity in each group.

Groups ^(1)^	MDA (nmol/mg Protein)	AChE (%mg Protein)
Normal	8.23 ± 0.13	101.86 ± 4.21
Control	16.17 ± 1.18 **	128.46 ± 2.19 *
Bio-RSE 50 mg/kg + Sco	14.99 ± 0.78	121.32 ± 2.23
Bio-RSE 100 mg/kg + Sco	12.93 ± 1.08 ^#^	116.54 ± 1.19 ^#^
Bio-RSE 200 mg/kg + Sco	10.91 ± 0.09 ^#^	108.65 ± 1.99 ^#^
DNPZ	13.69 ± 1.15 ^#^	115.68 ± 2.56 ^#^

^(1)^ Animal treatments are shown in Section 4. All values are the mean ± SD (*n* = 12/group). * *p* < 0.05 and ** *p* < 0.01 when compared to vehicle-treated normal group; ^#^ *p* < 0.05 when compared to the scopolamine-treated control group.

## Data Availability

Not applicable.

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
