# Peer review of "The Influence of Tyrosol-Enriched Rhodiola sachalinensis Extracts Bioconverted by the Mycelium of Bovista plumbe on Scopolamine-Induced Cognitive, Behavioral, and Physiological Responses in Mice"

_molecules, 2022, doi:10.3390/molecules27144455_

Round 1
Reviewer 1 Report
The authors of the manuscript ‘The influence of tyrosol-enriched Rhodiola sachalinensis extracts bioconverted by the mycelium of Bovista plumbe on scopolamine-induced cognitive, behavioral, and physiological responses in mice’ explored that the activation of the cholinergic system and the inhibition of oxidative stress of Bio-HKC on scopolamine (Sco)-induced memory impairment in mice.
There are several questions as below about this manuscript:
1. Since the authors detected the changes level of salidoside and tyrosol after bioconvertion of the Rhodiola sachalinensis extracts, I wonder the amount changes of other ingredients in the Rhodiola sachalinensis extracts.
2. The SOD and catalase levels of brain tissue should be provided.
3. The probe data should be displayed in Morris water maze in Fig 3.
4. It is better that the authors should indicate which kind of Aβ (Aβ1-40? or Aβ1-42?) was detected in the experiment.
5. The molecular weight should be provided in all Western blotting results.
6. The orientation is not marked in Fig 3C.
7. The authors should provide good quality representative images in Fig 1 and Fig 6.
8. Please check whole manuscript carefully since there are lots of format errors.
e.g. a full stop in red in line 84
missing full stop in line 90
It seems the authors used different fonts in whole manuscript.
Author Response
Reviewer 1
There are several questions as below about this manuscript:
- Since the authors detected the changes level of salidoside and tyrosol after bioconvertion of the Rhodiola sachalinensis extracts, I wonder the amount changes of other ingredients in the Rhodiola sachalinensis extracts.
Ans : We added
- The SOD and catalase levels of brain tissue should be provided.
Ans : We added
- The probe data should be displayed in Morris water maze in Fig 3.
Ans :
- It is better that the authors should indicate which kind of Aβ (Aβ1-40? or Aβ1-42?) was detected in the experiment.
Ans : We corrected
- The molecular weight should be provided in all Western blotting results.
Ans : We added
- Theorientation is not marked in Fig 3C.
Ans : We added
- The authors should provide good quality representative images in Fig 1 and Fig 6.
Ans : We corrected as much as we could.
- Please check whole manuscript carefully since there are lots of format errors.
e.g. a full stop in red in line 84
missing full stop in line 90
It seems the authors used different fonts in whole manuscript.
Ans : We corrected
Reviewer 2 Report
In this work, the authors investigated the anti-dementia effect of bioconverted R. sachalinensis ethanol extract (Bio-RSE) on scopolamine-induced cognitively impaired mice by examining the influence of the extract on memory and learning ability using Y-maze, water maze, and passive avoidance tests. The authors also evaluated the impact of Bio-RSE on the blood and brain antioxidant machinery of the memory-impaired mice as well their expression of beta amyloid peptide and amyloid precursor protein, implicated in the pathogenesis of Alzheimer's disease. Overall, this work is interesting and the authors made significant effort in the conceptualization and execution of the research objectives. Prior to publication, the work is still in need of considerable improvements. Comments and guidance are provided below for this purpose.
General
1. Salidroside and tyrosol are clearly the two major active ingredients in Bio-RSE. I recommend the authors determine the actual content of these compounds before and after bioconversion using quantitative HPLC analysis.
2. Considering that salidroside and tyrosol are the marker compounds for Bio-RSE, it highly plausible that the observed effects in this study are attributable to these compounds, or perhaps, not. To clarify the basis of the reported effects, I recommend a 7th and 8th group for this two compounds be included in the animal experiment.
3. The Methods section is bereft of details, especially how the data were obtained/calculated. This should be corrected.
4. It seems a bit odd that the authors evaluated the antioxidant activity (SOD, CAT) and lipid peroxidation (MDA) and expression of Nrf-2/HO-1, but did not evaluate the intracellular ROS. Is there any explanation for this?
5. The manuscript is replete with grammatical errors, poor phrasing, and word choice. Authors should consider engaging a professional English language editing service for improving the scientific writing.
Specific comments.
Abstract
Line 18: Physiological appears to be a typo. Pharmacological seems to be more appropriate.
Line 21: A typo with respect to the aim. Authors should replace Bio-HKC with Bio-RSE.
Authors should also clarify the meaning of Bio-RSE as well as Bio-HKC and use these abbreviations properly and consistently in the Abstract as well as the entire manuscript.
Introduction section
A considerable part of this work involved examining the impact of Bio-RSE on beta amyloid peptide and amyloid precursor protein. There was no mention of these proteins as well as their relevance to this study in the Introduction section. This should be corrected and corresponding citations included. I will suggest the authors also consider the physiological role of beta amyloid peptide as well as its pathological implications.
Results and Discussion
-Authors should provide an explanation pertaining to the observed increase in phenolic content after bioconversion of the ethanol extract.
-Authors report total phenolic content, flavonoid content and antioxidant activity of Bio-RSE in mcg/mL. Were this value based on the 1% solution or the entire crude extract.
-Authors should provide the actual values of salidoside and tyrosol in RSE before and after bioconversion.
-Table 5, statistical significance should be included in the table values.
-Figure 2, Statistical significance should be included on the bars.
-Figure 6. In Figure 6A and B, authors should consider providing images that include bands of the protein control.
Lines 334 - 339: Authors should clarify that the results being discussed are from a different study.
Materials and method
-Line 398: 'The R. sachalinensis ethanol extract was prepared as a 1% solution and then autoclaved.' Authors should provide a detailed description of the extract preparation starting from how the plant material was obtained, processed and extracted. How was the extract preserved?
-There is no mention of the HPLC method. Detail description should be provided.
-Although methods for the total phenolic content, flavonoid as well as antioxidant activity are fairly common, the details provided here are insufficient.
-Did the authors use any loading controls for the Western blotting experiments. Line 495, 'Equal amounts of prostate proteins...' How did authors determine protein amount. This should be included in the manuscript.
Author Response
Review 2
General
- Salidroside and tyrosol are clearly the two major active ingredients in Bio-RSE. I recommend the authors determine the actual content of these compounds before and after bioconversion using quantitative HPLC analysis.
Ans : We added as you said
- Considering that salidroside and tyrosol are the marker compounds for Bio-RSE, it highly plausible that the observed effects in this study are attributable to these compounds, or perhaps, not. To clarify the basis of the reported effects, I recommend a 7th and 8th group for this two compounds be included in the animal experiment.
Ans : As we mentioned in the introduction, RSE can be said that salidroside and tyrosol are the main substances. On the other hand, Rhodiola extract is currently used as a material that can help improve fatigue, and the indicator material is rosavin. In addition, it is known that there are many active substances of various terpenoids such as geraniol glucoside and tricin, and so on.
Bioconversion using the mycelium used in this study means that RSE is converted to BIO-RSE and also tyrosol is converted from salidroside. We described these as aglycon-enriched products in the introduction. I think it is also different from the tyrosol single substance.As you mentioned, tyrosol itself has already been shown to be effective in improving cognitive function in several groups. The main content of this study was to prepare Bio-RSE rich in tyrosol through bioconversion using mycelium rather than chemical synthesis, and that Bio-RSE exhibits high antioxidant activity due to its high polyphenol and flavonoid content.
- The Methods section is bereft of details, especially how the data were obtained/calculated. This should be corrected.
Ans : We Corrected
- It seems a bit odd that the authors evaluated the antioxidant activity (SOD, CAT) and lipid peroxidation (MDA) and expression of Nrf-2/HO-1, but did not evaluate the intracellular ROS. Is there any explanation for this?
Ans : We fully agree with your opinion. We thought of intracellular ROS in relation to mitochondrial biogenesis. Mitochondria are key cell organelles in that they are responsible for energy production and control many processes from signaling to cell death. Thus, mutations, environmental toxins and chronic ischaemic conditions could affect the mitochondrial redox balance and lead to the development of pathology.
We used scopolamine to inhibit cognitive function, which is known to impair memory by inducing oxidative stress to induce neuronal damage. Therefore, this study was intended to verify the factors related to oxidative stress including MDA and the effect of improving cognitive function through various behavioral analysis experiments.
However, if there is an opportunty, I would like to measure the metabolites and ATP related to mitochondrial biogenesis by referring to your opinion, and to investigate the decrease in ROS generated from the actual mitochondria.
- The manuscript is replete with grammatical errors, poor phrasing, and word choice. Authors should consider engaging a professional English language editing service for improving the scientific writing.
Ans : We Corrected
Specific comments.
Abstract
Line 18: Physiological appears to be a typo. Pharmacological seems to be more appropriate.
Ans : We Corrected
Line 21: A typo with respect to the aim. Authors should replace Bio-HKC with Bio-RSE.
Authors should also clarify the meaning of Bio-RSE as well as Bio-HKC and use these abbreviations properly and consistently in the Abstract as well as the entire manuscript.
Ans : We Corrected
Introduction section
A considerable part of this work involved examining the impact of Bio-RSE on beta amyloid peptide and amyloid precursor protein. There was no mention of these proteins as well as their relevance to this study in the Introduction section. This should be corrected and corresponding citations included. I will suggest the authors also consider the physiological role of beta amyloid peptide as well as its pathological implications.
Ans : We think your opinion is good too. Alzheimer's disease (AD) is a most common neurodegernative disorder affecting the aged popolation. The pathology of AD is characterized by senile plagues (predominantly consisting of aggregated β-amyloid) and intracellular neurofibrillary tangles (formed by tau aggregates). As we mentioned introduction, the toxicity of amyloid β (Aβ) is related to increased oxidative stress and lipid peroxidation in neurons. Hence, the overproduction of ROS contributes to the progression of Alzheimer disease (AD), which causes oxidative damage via a neurotoxicity mechanism and induces memory loss. Of course, the physiological contents of beta amyloid and amyloid precursor protein were not specifically mentioned. This is because the oxidative stress caused by scopolamine in the animal model we used, resulting in beta amyloid accumulation, was phenomenologically confirmed because it has already been reported.
Results and Discussion
-Authors should provide an explanation pertaining to the observed increase in phenolic content after bioconversion of the ethanol extract.
Ans : We added
-Authors report total phenolic content, flavonoid content and antioxidant activity of Bio-RSE in mcg/mL. Were this value based on the 1% solution or the entire crude extract.
Ans : We measured based on the entire crude extract.
-Authors should provide the actual values of salidoside and tyrosol in RSE before and after bioconversion.
Ans : We added
-Table 5, statistical significance should be included in the table values.
Ans : We added
-Figure 2, Statistical significance should be included on the bars.
Ans : We added
-Figure 6. In Figure 6A and B, authors should consider providing images that include bands of the protein control.
Ans : In general, β-actin or GAPDH is used as protein control, but the one we used in this study is HSC70 protein. There is no special reason. However, especially in animal experiments, HSC70 is easier to use than β-actin or GAPDH to be used as a house keeping gene. References on this are as follows:
References
TonEBP/NFAT5promotes obesity and insulin resistance by epigenetic suppression of white adipose tissue beiging. Lee HH, An SM, Ye BJ, Lee JH, Yoo EJ, Jeong GW, Kang HJ, Alfadda AA, Lim SW, Park J, Lee-Kwon W, Kim JB, Choi SY, Kwon HM. Nat Commun. 2019 10(1):3536.
Transcriptional RegulatorTonEBPMediates Oxidative Damages in Ischemic Kidney Injury.Yoo EJ, Lim SW, Kang HJ, Park H, Yoon S, Nam D, Sanada S, Kwon MJ, Lee-Kwon W, Choi SY, Kwon HM. Cells. 2019 (10):1284
Gamma Irradiated Rhodiola sachalinensis Extract Ameliorates Testosterone-Induced Benign Prostatic Hyperplasia by Downregulating 5-Alpha Reductase and Restoring Testosterone in Rats. Xin Q, Kwon MJ, Lee JW, Kim KS, Chen H, Campos MG, Tundis R, Cui CB, Cho YH, Cao H.Molecules. 2019 24(21):3981.
Lines 334 - 339: Authors should clarify that the results being discussed are from a different study.
Ans : We corrected
Materials and method
-Line 398: 'The R. sachalinensis ethanol extract was prepared as a 1% solution and then autoclaved.' Authors should provide a detailed description of the extract preparation starting from how the plant material was obtained, processed and extracted. How was the extract preserved?
Ans : We added. And both RSE and Bio-RSE are lyophilized and stored in a -80℃ deep freezer until use.
-There is no mention of the HPLC method. Detail description should be provided.
Ans : We added
-Although methods for the total phenolic content, flavonoid as well as antioxidant activity are fairly common, the details provided here are insufficient.
Ans : We added
-Did the authors use any loading controls for the Western blotting experiments. Line 495, 'Equal amounts of prostate proteins...' How did authors determine protein amount. This should be included in the manuscript.
Ans : We answered above. As we mentioned above, HSC70 is a universally used house keeping protein.
Round 2
Reviewer 1 Report
The authors have partially addressed my some comments. However, there are still some issues need to be improved.
1. It is better that the authors could draw a better graphic mechanism for understanding.
2. In the figure 1, the conventional sign is too small to see.
3. The probe data should be displayed in Morris water maze in Fig 3.
4. Although the authors indicated that Aβ1-42 was detected in the method part, they should change Aβ to Aβ1-42 in figure 5 as well as in the result 2.12.
5. Please check whole manuscript carefully since the authors used different fonts in whole manuscript.
Author Response
- It is better that the authors could draw a better graphic mechanism for understanding.
Ans : We added.
- In the figure 1, the conventional sign is too small to see.
Ans : We totally agree with your opinion. Since this HPLC histogram cannot be corrected on its own, it has been heavily adjusted to make it as visible as possible. We added quantitative results when we first make correction and supplement, so we hope you will understand.
- The probe data should be displayed in Morris water maze in Fig 3.
Ans : The result of the probe you mentioned is probably an experiment to see if the mice remember the lacation of the platform as it is after removing the platform. To be more precise, the probe trial is preformed to verify the animal's understanding of the platform location, and observe the strategy that the animal follows when it discovers the platform is not there. On the day of the last experiment, we also conducted that experiment. All experimental groups treated with scopolamine showed similar results to the previous results. However, there were some results that were somewhat difficult for us to interpret. We used ICR mice, not TG mice. Maybe that's why, when the last platform was removed, the normal group confirmed unexpected results. In the normal group, as on the first day, they played for over a minute as if they were enjoying swimming. We consulted several laboratories, but most answered that we had naver seen such a phenomenon with TG mice. We had no choice but to conclude that it may appear as a characteristic of ICR mice with various genetic backgrounds. Nevertheless, if you ask us to add the result again, we will.
- Although the authors indicated that Aβ1-42 was detected in the method part, they should change Aβ to Aβ1-42 in figure 5 as well as in the result 2.12.
Ans : We corrected.
- Please check whole manuscript carefully since the authors used different fonts in whole manuscript.
Ans : We are really sorry. But we are not sure exactly what you mean by different fonts. We continue to make corrections and supplements using the files received after submission. If you explain in more detail, we will re-edit it again. Please.
Reviewer 2 Report
I am satisfied with the authors' response and improvements made in the manuscript.
Author Response
Thank you so much.
